# Vegetation–Lake–Sand Landscape of Northeast China Sandy Land between 1980 and 2022: Pattern, Evolution, and Driving Forces

## Weiyi Lu [1], Geer Teni [2] and Huishi Du [1,*]

1   College of Geographical Science and Tourism, Jilin Normal University, Siping 136000, China; lwy160420@163.com
2   Degree Programs in Life and Earth Sciences, University of Tsukuba, Tsukuba 305-8577, Japan; tenigeer0212@gmail.com
*   Correspondence: duhs@jlnu.edu.cn

**Abstract:** Northeast China's sandy region is an arid and semi-arid zone highly susceptible to climate change. Investigating the long-term changes in the Northeast China sandy land (Northeast China sandy land, DBSL) landscape can provide an important basis for the ecological restoration of this region. This study analyzed long-term remote sensing data of the DBSL from 1980 to 2022 and explored the spatial pattern, evolution, and driving mechanisms. In 2022, vegetation was mainly distributed in the northwest, center, and southwest, covering a total area of 30,508.82 km$^2$. Areas with high and medium vegetation cover showed strong aggregation characteristics and were mainly distributed in the southwest, whereas those with low vegetation coverage were highly dispersed and widely distributed in the central region. Lakes were widely distributed in the northwest and central regions, with a total area of 2736.43 km$^2$. In the last 42 years, the vegetation cover decreased by 24.48%. Areas with high and medium vegetation coverage decreased in size, and those with low vegetation coverage first increased and then decreased, with overall decreases of 35.35%, 19.16%, and 6.88%, respectively. The overall area of the DBSL showed various degrees of degradation. Shrinking and dry lakes were concentrated in the sandy hinterland. The lake landscape changed significantly from 1990 to 2010, with a decrease in lake area of 27.41%. In contrast, the sandy area increased by 25.65%, indicating a high degree of desertification. However, from 2005 to 2022, desertification decelerated. The most important factors driving the evolution of the DBSL were socio-economic factors. The increase in human disturbance will have a certain impact on the landscape changes in the region in the short term. The national policy of returning farmland to fields and grasslands will affect the increase of vegetation and lake landscape area in the short term, and the sand area and excessive animal husbandry will be reduced. This study provides a scientific basis for ecological restoration and sustainable development in Northeast China.

**Keywords:** vegetation cover; lake wetlands; evolution; sandy land; DBSL

## 1. Introduction

Northeast China is an arid and semi-arid region with fragile ecosystems, making it highly susceptible to climate change [1–3]. Vegetation, lakes, and sandy land interact with each other, forming a composite ecosystem and shaping the landscape of this region [4]. At the beginning of this century, scientists started to investigate the interaction between plant growth and sandstorm activities in arid and semi-arid areas [5], focusing on sand-vegetation and wind–lake interactions. Liu et al. found that the formation of sand on the east coast of Qinghai Lake was related to the decline of the lake. The surface of the lake shrank, and the exposed sediments on the bottom of the lake were eroded by the west wind and then accumulated to form the sand on the east bank [6,7]. Vegetation cover is the main factor controlling wind erosion, significantly impacting wind speed and

wind–sand flow [8]. A reduction in vegetation cover may lead to irreversible climate change. In contrast, increasing the vegetation cover could reduce wind speed and the wind–sand transport flux, thus reducing surface wind erosion. Wassim used the sampling method to investigate the vegetation situation under different conditions in southern Tunisia. The main research results showed that the vegetation coverage was related to climatic conditions [9]. Vegetation protects the surface by covering it and capturing sand particles. When the vegetation cover is reduced, sand sedimentation and erosion increase, and sand flow decreases exponentially with the vegetation coverage rate [10]. With sand transport, fine particles provide nutrients and water for plant colonization and growth. Zhou et al. used long-term data from China's semi-arid area research station to study the interaction between sand fixation and sand vegetation growth and pointed out that the accumulation of soil nutrients and vegetation biomass increases sand and vegetation cover areas [11], promoting changes in the vegetation landscape [12].

Lakes are an important surface water resource and are crucial in sandy areas. They play an indispensable role in maintaining the ecological and environmental balance of arid and semi-arid areas [13]. The expansion and shrinkage of lakes affect regional vegetation habitats and determine the distribution pattern of sandy vegetation [14]. Lakes provide conditions for vegetation growth and survival and have an important impact on sand transport [15]. Chen et al. [16] evaluated the interrelationship between vegetation and lakes on the Qinghai-Tibet Plateau and found that grassland was significantly negatively correlated with lake expansion [17]. Chun et al. [18] reported that, along with the increase in desertification in the Xilingol area, the Darinur Lake and Hulun Lake surface areas gradually decreased [19]. Du et al. [20] mentioned in their study the distribution of wind–sand–lake–vegetation landscape patterns in the Songnen sandy lands. Although, in China, the interaction among vegetation, lakes, and sandy landscapes has been extensively studied, the relationship among vegetation, lakes, and sandy areas is still largely unclear. This makes it necessary to determine the changes in vegetation–lake–sand landscape patterns in arid and semi-arid areas and reveal the underlying mechanisms.

The DBSL has both arid and semi-arid characteristics [21]. In terms of vegetation, the normalized difference vegetation index (NDVI) is significantly positively correlated with average annual precipitation and temperature [22]. Climate warming reduces the moisture content in the soil, which inhibits vegetation growth and increases the likeliness of drought events [23], ultimately leading to the expansion of sandy areas [24]. In recent years, China's economy has developed rapidly, and human high-intensity land use, including over-planting, overgrazing, and excessive groundwater extraction, has had a profound impact on the ecological environment [25]. Shen et al. [17] pointed out that the increase in the desert area and inhibited vegetation growth in Inner Mongolia are mainly due to the poor management of water resources and unsustainable land use as a result of rapid population growth; this is further exacerbated by rising temperatures [26,27].

Spatial analysis, the MEDALUS model [28], the residual trend method [29], and other means have been employed to investigate the comprehensive impacts of vegetation, sandy landscape changes, climate factors, and human activities on landscape evolution [30]. Although vegetation and sandy landscapes are more sensitive to climate factors, human activities reduce this sensitivity. In this sense, considering the combined impacts of climate and human factors, this study used gray correlation analysis to quantitatively analyze the driving force of vegetation–lake–sand landscape evolution. This novel statistical method can detect spatial heterogeneity and reveal underlying mechanisms [31]. Its advantage is the quantitative analysis of driving factors [32], which increases our understanding of the effects of climate and human activities on the landscape.

In recent years, the strategic role of the sandy area in Northeast China has become evident [33–35]. This region is crucial for maintaining and stabilizing regional, ecological, food, and climate security [36]. The spatial overlap between the sandy land in Northeast China and the interlaced areas of China's agricultural and animal husbandry production activities has resulted in fragile areas that are highly sensitive to climatic changes [37–39]. In

view of this, using the DBSL as the study area, long-term sequence satellite image data and geographical detector model data from the Google Earth Engine (GEE) cloud platform were obtained to determine the spatial and temporal dynamic evolution, annual change trend, and driving forces of the vegetation–lake–sand landscape pattern. The results provide a reference for the restoration of the ecological environment in Northeast China.

## 2. Materials and Methods

### 2.1. Study Area

The DBSL is distributed in the arid, semi-arid, and semi-humid areas between 42°31′–50°37′ N latitude and 115°31′–129°12′ E longitude, including Horqin sandy land, Songnen sandy land (SSL), and Hulunbuir sandy land, with a total area of $5.166 \times 10^5$ km² distributed in Inner Mongolia Autonomous Region, Liaoning Province, Jilin Province, and parts of Heilongjiang Province (Figure 1). This study was based on the extent of the Horqin sandy land determined by the water basin segmentation method to study the sandy land in the northeast region [20]. The DBSL sandy land area is located in the transition zone of the Inner Mongolia Plateau and the Northeast Plain and the temperate monsoon climate zone in eastern Asia. As a farming-pastoral zone, it is sensitive to climate change. The area is mainly affected by the polar vortex and subtropical high pressure. Northwest winds are common, and precipitation changes from southeast to northwest. Annual precipitation is low, albeit with large variability. Rainfall is unevenly seasonally distributed and concentrated from June to August; annual precipitation is 310–450 mm, with an annual evaporation of 1400–1800 mm. The soil belongs to the temperate grassland black-land belt, and the dominant soil types are black calcium soil, wind sand soil, saline and alkaline soil, and swamp soil. The main vegetation area is the meadow grassland area of Songliao Plain, dominated by *Leymus chinensis*, *Filifolium sibiricum*, *Arundinella hirta*, *Stipa baicalensis*, *Hemarthria japonica*, *Calamagrostis brachytricha*, *Salix mongolica*, and *Phragmites australis* (Cav.). The region is an important animal husbandry base and commodity grain base. In 2022, the GDP of Northeast China was CNY 11,205.3 billion, the total grain output was 32,489.34 t, and the number of large livestock at the end of the year was 8,563,700. The region is characterized by multi-ethnic settlements and the formation of characteristic cultural industrial branches [40,41].

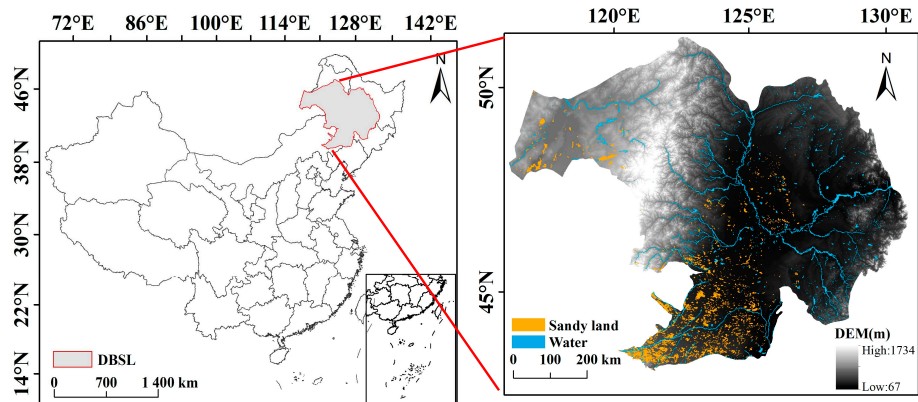

**Figure 1.** Location of the study area. Note: The map is based on the standard map of GS (2022) 4307, the standard map service website of the National Bureau of Surveying, Mapping and Geographic Information, and the base map has not been modified.

### 2.2. Materials

#### 2.2.1. Remote Sensing Data

Using the GEE cloud platform, Landsat time series satellite image data were obtained for 1980, 1985, 1990, 1995, 2000, 2005, 2010, and 2022 for the peak vegetation growth from mid-August to mid-September. DEM (digital elevation model) data were obtained from the geographical special data cloud and had a spatial resolution of 250 m. ArcGIS10.8

software was employed to mosaic, splice, crop, and process elevation data [42]. The natural break-point method was used to reclassify the data. Using the object-oriented method, sand and lake information was extracted to enable the full use of spectral information, geometry, texture characteristics, and the relationship between sand and other objects in the multi-phase image. A segmentation scale of 200 was applied for multi-resolution segmentation [43]. Since most of the images of the Landsat series were mixed image elements, the shape of the object was blurred, and the shape characteristics had less impact on image segmentation. Therefore, the weight parameters of shape heterogeneity shape and spectral heterogeneity color were set to 0.1 and 0.5, respectively. Each band participated equally in sand information extraction. The weight value of each band was set to 1 [44]. Based on preliminary research and field verification, eCognition9.0 software was used to complete the division of sand and lake information. The overall classification accuracy was 93%, with a Kappa coefficient of 0.98.

### 2.2.2. Meteorological Data

Temperature and precipitation are the two most basic climatic elements that reflect regional climate characteristics. The temperature and precipitation data used in this study were obtained from the China Meteorological Data Network (http://data.cma.cn/ (accessed on 27 February 2024)), using meteorological stations in Harbin City, Tieli City, Qiqihar City, Shangzhi City, Hailun City, Tailai County, Fuyu County, Tonghe County, Keshan County in Heilongjiang Province, Changchun City, Fuyu City, Baicheng City, Qian a'n County, Qianguoer Rose Mongolian Autonomous County, Tongyu County, Changling County in Jilin Province, and Zhalantun City in Inner Mongolia. Average annual wind speed, annual precipitation, and average annual temperature data from 1980 to 2022 were considered. In data processing, sites missing data were removed and the missing data were interpolated. The sites are evenly distributed within and around the region, reflecting the overall situation.

### 2.2.3. Statistical Data

The selected data indicators were total population, China's GDP, cultivated land area, and per capita GDP data of the study area from 1980 to 2022 (Table A1). These indicators can adequately characterize the socio-economic status of a region (socio-economic data from the Inner Mongolia Statistical Yearbook, Jilin Statistical Yearbook, Liaoning Statistical Yearbook, Heilongjiang Statistical Yearbook, and Bureau of Statistics) (http://www.cnki.net (accessed on 27 February 2024)) [45].

### 2.3. Methods

#### 2.3.1. Calculation of Vegetation Coverage

The scope of sand and lake interpretation was removed by masking, vegetation coverage was calculated using the mixed image meta-decomposition method in unclassified areas, and the vegetation index obtained after normalization of the original data was used to reverse the vegetation coverage information. The following equation was used:

$$F = \left[ (NDVI - NDVI_{soil}) / (NDVI_{veg} - NDVI_{soil}) \right]^2 \tag{1}$$

where FVC is the vegetation coverage, $NDVI_{soil}$ is the $NDVI$ value of the bare soil or vegetation-free area, and $NDVI_{veg}$ represents the $NDVI$ value of the image element completely covered by vegetation, that is, the $NDVI$ value of the pure vegetation image element.

For most bare land, the $NDVI_{soil}$ value should theoretically be close to 0; at full vegetation coverage, the $NDVI_{veg}$ value is close to 1. Affected by various natural conditions such as time and region, the $NDVI$ needs to be used to determine the $NDVI_{soil}$ and $NDVI_{veg}$ values of different images. By analyzing the $NDVI$ data of Landsat series images, combined with the actual situation of the pine sand and the cumulative probability distribution table of the $NDVI$ values, the confidence value was 0.5% [46].

2.3.2. Gray Correlation Degree

The gray system theory is generally applied to analyze small datasets and when information is lacking. Its main feature is that it can use a small amount of data for simulation and estimation and infer unknown information from known information [47]. This method uses the association theory of the gray system and builds a gray association model to find the gray correlation degree of each feature sequence and the parent sequence. The greater the correlation, the closer the relationship between the comparison sequence and the reference sequence studied in the system. The basic principle of this method is to determine whether the two objects are related according to the similarity of the geometry of the object. The closer the connection, the more similar the geometry of the curve, and the higher the gray correlation [48].

Let there be a total of $x$ known samples, each of which corresponds to $y$ characteristic factors. The vector of the characteristic factor of the $j$ sample is expressed as $tj = (t_{1j}, t_{2j}, \ldots, t_{yj})$, and the $y$ characteristic factors of $x$ known samples are represented by matrix $T = (t_{ij})_{yx}$, where $t_{ij}$ represents the $i$ characteristic factor of the $j$ sample, $i = 1, 2, \ldots, y; j = 1, 2, \ldots, x$:

$$\xi'_{kj}(i) = \frac{\min\limits_{j}\min\limits_{i}|r_k(i) - r_j(i)| + \rho\max\limits_{j}\max\limits_{i}|r_k(i) - r_j(i)|}{|r_k(i) - r_k(i)| + \rho\max\limits_{j}\max\limits_{i}|r_k(i) - r_j(i)|} \tag{2}$$

In this equation, the correlation coefficient of the $k$ sample to be identified and the $k$ known sample at the $k$ characteristic factor, $\min\limits_{j}\min\limits_{i}|r_k(i) - r_j(i)|$, is the minimum difference between the two levels, $\max\limits_{j}\max\limits_{i}|r_k(i) - r_j(i)|$ is the maximum difference between the two levels, and $\rho$ is the resolution coefficient, generally 0.5, $k = 1, 2, \ldots, x$. Combining the correlation coefficients of each point, $\alpha'_{kj}$ degrees of correlation between the $k$ sample to be identified and the $j$ known sample can be obtained. The equation is as follows:

$$\alpha'_{kj} = \frac{1}{y}\sum_{i=1}^{y}\xi'_{kj}(i) \tag{3}$$

The above equation can be used to calculate the correlation between the sample to be identified and all known samples, and the known sample most closely related to the sample to be identified can be determined according to the principle of maximum correlation.

2.3.3. Principal Component Analysis

Principal component analysis (PCA) is a multivariate statistical analysis method [49]. Based on the idea of dimension reduction, this method transforms the original multiple variables into a few comprehensive variables through linear transformation. These variables are not related to each other and can reflect most of the information of the original variables. At the same time, the information components do not overlap each other. In order to further study the role of driving factors on landscape change, this study analyzed the main components of the driving factors affecting the landscape change of the northeast sandy land and then found the main driving factor to provide a certain basis for the subsequent ecological protection of the northeast sand.

**3. Results**

*3.1. DBSL Vegetation–Lake–Sand Landscape Pattern Status*

In 2022, the vegetation in the DBSL was mainly distributed in the central, northwestern, and southwestern parts, with a small distribution area in the central region, with a total area of 30,508.82 km$^2$, accounting for 34.52%, 46.83%, and 18.65% (Figure 2), respectively. From the perspective of spatial distribution, areas with high and medium vegetation coverage were significantly distributed across the study region. The DBSL was dominated by areas

with high and medium vegetation coverage. Areas with high vegetation coverage were mainly distributed in the south and north of the DBSL, with a few areas in the central region, covering a total of 10,533.48 km². The area with medium vegetation coverage was the largest one and was concentrated in the southwest and northwest of the DBSL, covering an area of 10,779.47 km². The area with low vegetation coverage was widely distributed in the northwest of the DBSL and adjacent to the sandy area, covering 5691.0213 km². Presently, the DBSL lake landscape is concentrated in the middle and lower parts as well as the southwest, with small lakes scattered in the northwest of the research area. The total lake area covers 2667.554 km². Generally, the area covered by vegetation accounts for most of the studied area. The sand areas of the DBSL are mainly scattered throughout the southern part and concentrated in the northwest, with some small areas in the central part. In total, the sand area covers 10,546.07 km².

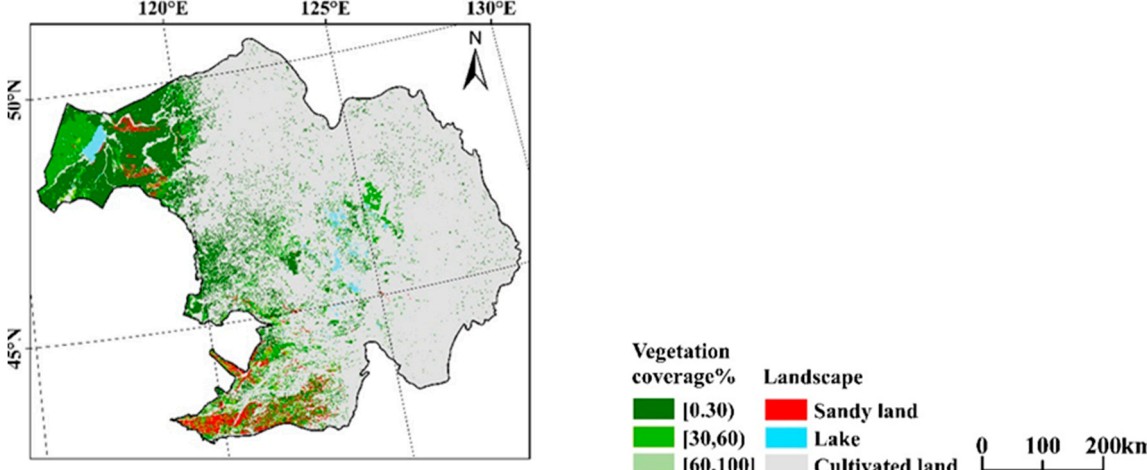

**Figure 2.** Current status of vegetation–lake–sand landscape pattern in DBSL.

### 3.2. Spatial Differentiation of the DBSL Vegetation–Lake–Sand Landscape Pattern

3.2.1. Temporal Changes in the DBSL Vegetation–Lake–Sand Landscape Pattern

Over the past 42 years, the vegetation area of the DBSL has decreased constantly. In 1980, the DBSL vegetation landscape was widely distributed in the northwest of the research area, with a total area of 40,402.2 km² (Figure 3). The high vegetation coverage area was concentrated in the northwest and the medium vegetation coverage area in the southwest. From 1980 to 2005, over reclaimed farmland in this area, vegetation degradation was severe, with a decrease in vegetation coverage of 9.57%. In particular, the highly vegetated area was seriously degraded (Figure 4). Consequently, the areas with medium and low vegetation coverage increased first and then decreased. Among them, the vegetation coverage area of Horqin sandy land showed a large degree of reduction (−8.92%), and the high vegetation coverage area significantly decreased by 7.10%. The vegetation area of Hulunbuir increased by 0.13%. In 2005, the grassland protection policy in Inner Mongolia was implemented, and grassland reduction decreased within 5 years, with a decrease in the vegetation area of 12.3% from 2005 to 2010. From 2010 to 2022, vegetation degradation was effectively controlled, with a decrease in vegetation area of 1.45%. Overall, the area of DBSL vegetation landscape continues to decrease, with a total vegetation landscape area of 9893.4 km²; the DBSL vegetation coverage area is concentrated in the west, the vegetation landscape in the east and north is greatly reduced, and the high-coverage vegetation landscape area in the east is basically degraded.

The DBSL sand landscape is mainly distributed in the south and north of the research area. From 1980 to 2022, the overall lake area of the DBSL decreased. Taking 1990 as the dividing point, the area of lakes in the central, southwest, and northwest of the 1980–1990 research area expanded to varying degrees. Among them, the area of lakes in the SSL in the central part of the research area increased significantly by 12.47%. From

1990 to 2010, the lake area shrank significantly by 27.41%. During this period, the area of Hulunbuir lakes decreased after increasing, showing a fluctuating downward trend, with an overall decrease of 4.7%. Overall, throughout the 42 years, the lakes in the DBSL showed a shrinking trend. From 2000 to 2010, the area of the SSL and Horqin sandy lake clusters decreased sharply, resulting in a reduction in the overall lake landscape of the DBSL. From 2010 to 2022, the lake shrinkage trend improved, and the area decreased by 815.84 km². 

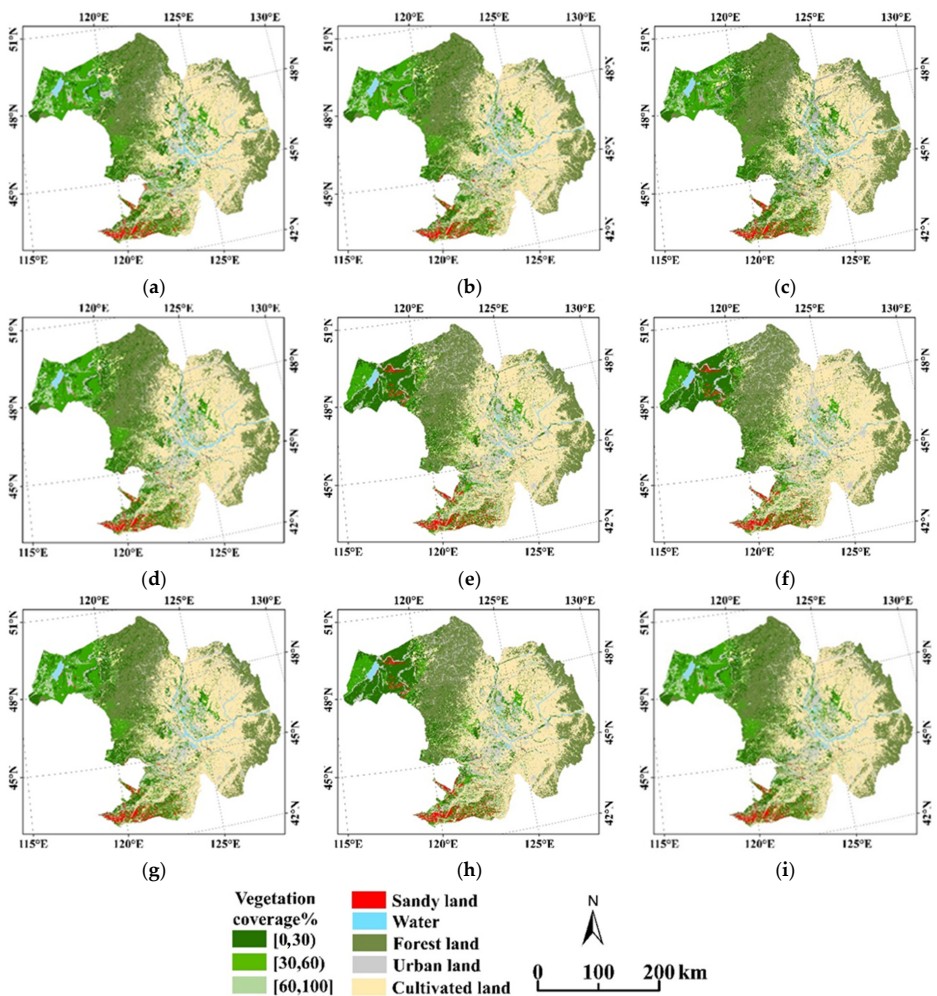

**Figure 3.** DBSL landscape evolution 1980–2022. (**a**) 1980 (**b**) 1985 (**c**) 1990 (**d**) 1995 (**e**) 2000 (**f**) 2005 (**g**) 2010 (**h**) 2015 (**i**) 2022.

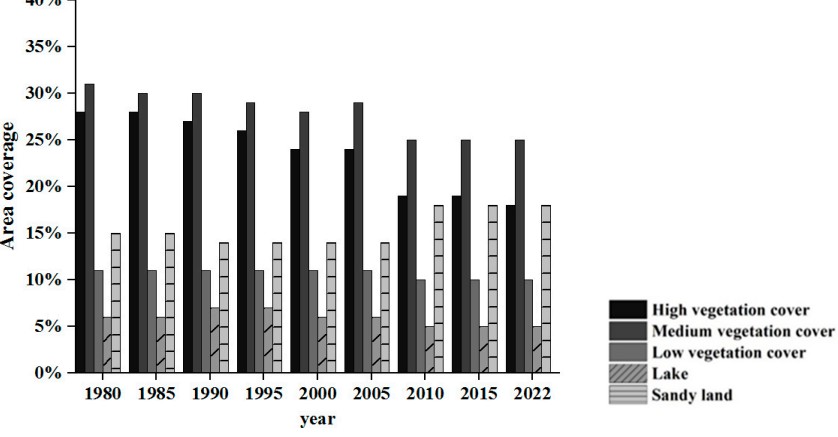

**Figure 4.** Statistical map of vegetation–lake–sand landscape changes in DBSL.

From 1980 to 2022, the sand area of the DBSL showed a trend of decreasing first and then increasing. Taking 2000 as the dividing line, from 1980 to 1995, the sandy area in Northeast China was relatively small and mainly located in the northern and southwestern parts. In 1995, the sandy area in the southwest decreased. The center of gravity of some sandy land in the middle of the research area moved to the southwest of the research area. The transformation of sand landscapes into arable land in the central area of the DBSL led to a sharp decline of 97.45%. In 1995–2000, Inner Mongolia implemented a program to convert farmland to forests. As a result, the ecological environment was improved, the sandy area was reduced again, and the trend of sand landscape expansion in the southwest of the research area was curbed. During this period, the overall area of the DBSL sand landscape was greatly reduced. From 2005 to 2010, the Hulunbuir area was overgrazed, the grassland was greatly degraded, and sand erosion was serious, resulting in a sharp increase in the sand area and substantial desertification, with an increase in the sand area of 16.25%. In 2010, the government implemented several policies for the establishment of nature reserves, the strengthening of ecological construction, and the effective control of desertification. In 2022, sand areas were less distributed in the central area of the DBSL. The sandy area in the northeast of Hulun Lake in the northwest decreased, with an overall decrease in sandy areas, and desertification slowed down. Overall, in nearly 42 years, the sandy area in Northeast China increased from 8393.139 km$^2$ in 1980 to 10,546.07 km$^2$ in 2022, with an increase of 25.65%. From 1980 to 2000, the sandy area decreased by 3.78%, mainly due to the reduction in the sandy area in the central and southwestern parts. From 2000 to 2022, the sandy area increased by 2469.92 km$^2$. During this period, desertification in the northwest intensified, resulting in a sharp increase in the sandy area, whereas in the central and northwestern regions, desertification was controlled.

3.2.2. Vegetation–Lake–Sand Landscape Interaction

The sandy area was highly correlated with the vegetation area; the correlation coefficient between the sandy area and the highly vegetated area was 0.7562, whereas that between the sandy area and the medium vegetation coverage area was 0.7128 (Figure 5); for the low vegetation coverage area, this value was 0.7013. There was a significant negative correlation between the sandy area and the medium and low vegetation coverage areas, with a correlation coefficient of −0.97. The size of the surface vegetation coverage area determines the intensity of wind erosion, and the sand wind erosion rate increases with the decrease in the vegetation coverage area. Vegetation coverage is the main factor controlling wind erosion as it has a significant impact on wind speed and surface sand deposition. Increasing the density of vegetation coverage can reduce wind speed and reduce wind–sand transport flux, thus reducing surface wind erosion. Therefore, as the vegetation cover increases, along with surface roughness, the accumulation of wind and sand on the surface will tend to stabilize. In this sense, increasing the vegetation coverage area is the focus of vegetation restoration in arid and semi-arid areas. The lake area was positively correlated with the vegetation coverage area, and the correlation with the low vegetation coverage area was high, with a correlation coefficient of 0.85. With lake degradation, the diversity of lake vegetation declines, water biodiversity is reduced, and the overall lake area is reduced. The correlation coefficient for the lake area and the high vegetation coverage area and medium vegetation coverage area were 0.74 and 0.71, respectively. The sand and lake area were negatively correlated, and the correlation between the two was 0.4316, which was lower than that between the lake and vegetation landscape and the sand and vegetation landscape, and the correlation coefficient is −0.87. The lake group was mainly distributed near the sand cover area of the research area, which formed the feature of mosaic distribution with the sand. The dynamic balance between the lake group and the sand area showed a long and dynamic balance.

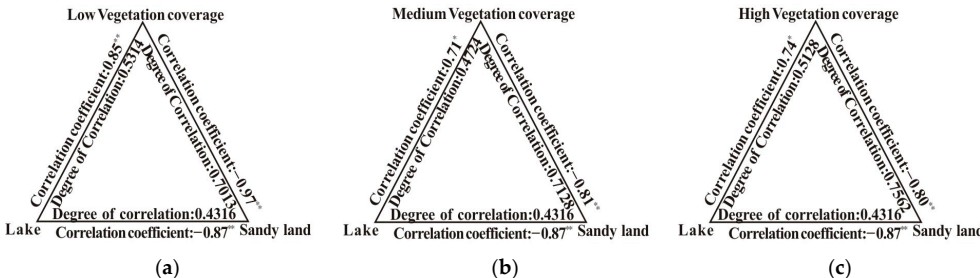

**Figure 5.** DBSL vegetation–lake–sand landscape correlations. (**a**) Vegetation, lake, and high vegetation cover correlations. (**b**) Vegetation, lake, and medium vegetation cover correlations. (**c**) Vegetation, lake, and low vegetation cover correlations. Note: * indicates passing the significance test of 0.05, ** indicates passing the significance test of 0.01.

### 3.3. Driving Mechanism of Vegetation–Lake–Sand Landscape Evolution

### 3.3.1. Climate Factors

Throughout the experimental period, wind speed and annual average temperature showed a fluctuating downward trend, whereas the annual average precipitation showed a fluctuating upward trend (Figure 6). The overall trend of the annual average evaporation was not obvious, but it slightly increased. Between 1980 and 2022, the temperature of the DBSL fluctuated greatly from year to year. After 2005, the temperature decreased rapidly by 6.13 °C/10 a; with the lowest temperature in 2006, the grain value appeared. After 2010, the growth rate was high, with an increase of 6.55 °C, and the temperature change tended to stabilize from 2010 to 2022. Overall, evaporation showed an upward trend, peaking in 1993 and 2005. The average annual precipitation in the research area fluctuated and was highest in 1998. The period before 1998 was relatively rainy, whereas, after 1998, precipitation decreased sharply. In 2001, it reached the lowest level, followed by a gradual increase. The climate change trend in the area fluctuated between dry and wet. The average wind speed fluctuated only slightly from 1980 to 2010 and was stable thereafter. However, after 2001, it slightly decreased. Generally speaking, the wind speed has decreased and the sand activity is relatively weak. Evaporation is an important indicator to measure the degree of drought in sand. Generally, at high temperatures, humidity is low, the wind speed is high, the air pressure is low, and evaporation is great. The annual evaporation in the research area fluctuated greatly, albeit with a decrease since 2000, indicating that the natural ecological conditions in the area have reversed and that the degree of drought has decreased.

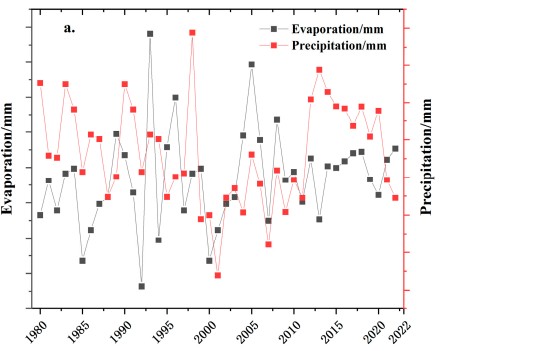 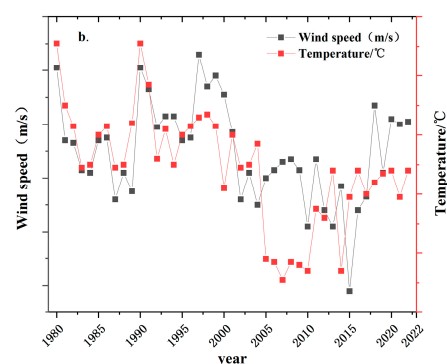

**Figure 6.** DBSL meteorological elements statistics. (**a**) Average annual evaporation vs. average annual precipitation statistics. (**b**) Average annual wind speed vs. average annual temperature statistics.

### 3.3.2. Socio-Economic Factors

The total population, GDP, arable land area, and number of large livestock increased toward the end of the experimental period (Figure 7). From 1980 to 2022, the total population in the research area showed a growth rate of 45.33%. Along with this, the regional

economy also developed rapidly. From 1980 to 2022, the DBSL regional GDP increased exponentially by 61.67%. Its changing trend is in line with the average income of farmers and herdsmen, which increased from 472.80 to 688.33 CNY/person. Throughout the research period, the overall arable land area also showed an annual growth trend. From 1980 to 1990, the growth trend of arable land area was relatively slow, with an increase of 44.4%. From 1990 to 2010, the growth rate accelerated, with an increase of 110.5%. Later, the government implemented the policy of returning farmland to forests and grassland, and the area of arable land decreased significantly from 2013 onward, with a total increase throughout the study period of 205.58 km$^2$. Grain production increased fluctuatingly, with an overall increase of $3.52 \times 10^9$ t. At the end of the year, the number of large livestock also showed an increasing trend. In 1980–2000, the growth trend of the number of livestock was flat, with a variation of 12.21%, whereas, in 2000–2022, the number of large livestock increased significantly by 8291 million. This has increased the carrying capacity of grassland and affected the vegetation landscape. Affected by the implementation of policies since 2000, and driven by policy and economic guidance, Northeast China has achieved intensive planting and agricultural and animal husbandry development, and human activities have had a certain impact on the regional sand landscape.

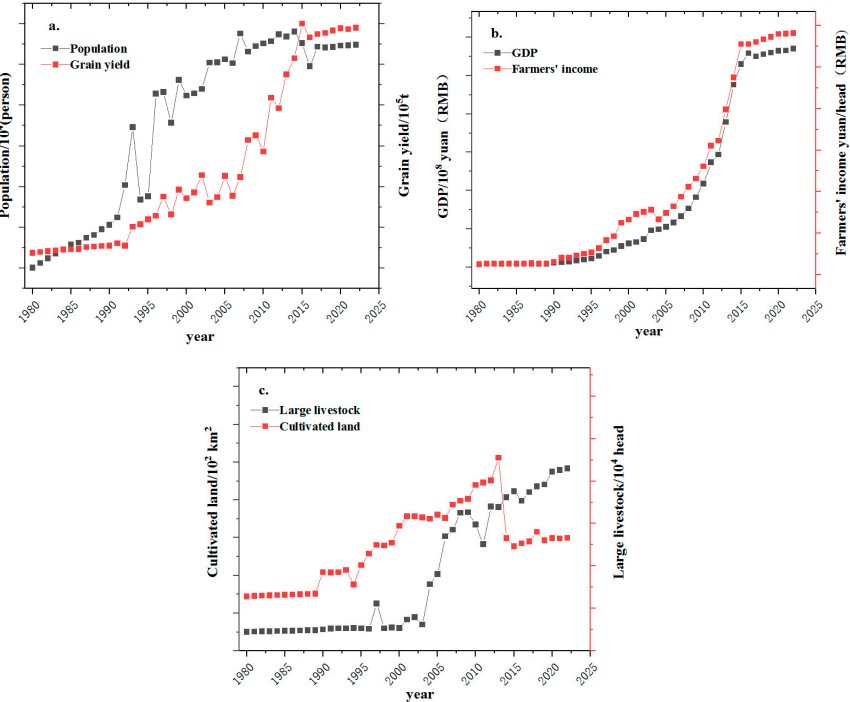

**Figure 7.** Statistical charts of socio-economic factors in DBSL. (**a**) Population vs. food production statistics. (**b**) GDP vs. per capita income of farmers and herdsmen. (**c**) Number of large livestock vs. cultivated area at the end of the year.

### 3.3.3. Quantitative Analysis of the Driving Factors of Vegetation–Lake–Sand Landscape Evolution

To further clarify the influencing factors of DBSL vegetation–lake–sand landscape evolution and analyze the role and effectiveness of natural and socio-economic factors, principal component analysis was adopted. Principal component analysis is a statistical analysis method, and the analysis results of this method are described by several inter-related quantitative dependent variables. The first principal component (Table A2), with a contribution rate of 54.45%, had a high loading of human factors, with a contribution rate of 18.92%; the second principal component had a high loading of natural factors; and the third principal component, with a contribution rate of 10.67%, had a high loading of human factors. Therefore, from a statistical point of view, socio-economic factors have a heavy impact on the vegetation–lake–sand landscape of the DBSL.

### 3.3.4. Relationship between Vegetation–Lake–Sand Landscape Evolution and Driving Factors

The DBSL vegetation coverage area was positively correlated with the total number of large livestock at the end of the year, average annual wind speed, and precipitation and negatively correlated with the cultivated land area, GDP, and total population (Figure 8). In recent years, the population has shown a fluctuating trend. The rapid economic development of the region has a two-sided impact on the ecosystem. The promulgation of national policies, such as the policy of returning farmland to grassland and forests, will cause changes in the land cover in the northeast region in the short term, increasing the area of vegetation landscape, and excessive animal husbandry and the reclamation of farmland will reduce the area of vegetation landscape. There was a significant negative correlation between the change in the vegetation coverage area and the area of cultivated land as well as the number of large livestock at the end of the year, with correlation coefficients of $-0.96$ and $-0.94$, respectively. Excessive animal husbandry and unreasonable farmland reclamation caused large-scale damage to vegetated areas. When the area of cultivated land and the number of large livestock at the end of the year increased, the vegetation area decreased significantly. The correlation coefficient between the average annual precipitation and the vegetation coverage area was 0.55. The DBSL precipitation increased, soil water stress increased, vegetation habitat quality increased, and the vegetation landscape area increased. The lake area was positively correlated with the annual average temperature but negatively correlated with the annual average evaporation and the cultivated land area. Among them, the positive correlation between annual average temperature and lake area was higher, with a correlation coefficient of 0.77. Overall, the average annual evaporation has increased, the lake landscape is reduced, and the expansion of cropland poses a threat to the development of small lakes; the increase in water consumption for growing crops also has an impact on the lake landscape, which reduces the area of lake landscape. The area of sandy land was positively correlated with annual average precipitation, total population, GDP, the per capita net income of farmers and herdsmen, the number of large livestock, and grain yield and negatively correlated with annual average temperature, annual average wind speed, and annual average evaporation. There was a significant positive correlation between the sandy landscape and the total population, and the correlation coefficient was 0.94. When the population in the DBSL region increases, the disturbance caused by human activities in the region also increases, the area of cultivated land expands, food production increases, and land desertification increases. The correlation coefficient between annual average wind speed and sandy land area was $-0.86$ and that between average wind speed and average precipitation was $-0.41$. When wind speed and precipitation increase, the sandy land area decreases. The precipitation increases, the soil humidity increases, the vegetation increases the ability to fix the sand, and the sand landscape decreases. In summary, the vegetation–lake–sand landscape changes in Northeast China are closely related to climate change and human activities. In this context, the fluctuation of annual precipitation decreases, the fluctuation of annual average temperature increases, potential evaporation increases, and the climate becomes colder and wetter, which leads to the expansion of the sandy land area. In addition, the total population in the region fluctuates more significantly, the demand for food continues to increase, and the area of cultivated land continues to expand. Short-term, high-intensity human activities can accelerate or delay the development of sandy land, lakes, and vegetation. Based on the results of this study, population pressure is too great, and the excessive reclamation of land will result in more pronounced desertification.

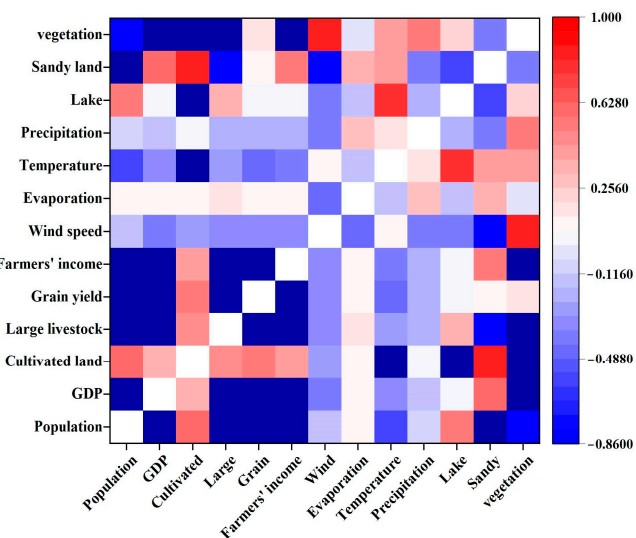

**Figure 8.** Correlation analysis between sandy land area, vegetation area, lake area, and climate factors and human factors.

## 4. Discussion

### 4.1. Dynamic Change Relationship of the Vegetation–Lake–Sand Landscape

There is a dynamic equilibrium relationship between vegetation–lake–sand landscape changes in the DBSL. When sand is transported by wind, fine particles provide nutrients and water for plant colonization and growth [50,51]. When plants grow to a certain extent, they affect the movement of wind-blown sand by covering the surface, intercepting surface sand, and dispersing airflow kinetic energy. The interaction between wind-blown sand and vegetation is obvious. Over the past 42 years, the DBSL vegetation coverage area showed a downward trend, while the sandy land area showed a fluctuating increasing trend. During this period, most of the high vegetation coverage areas were transformed into medium and low vegetation coverage areas. Some low vegetation coverage areas were reclaimed as cultivated land or degraded into bare land. The large-scale degradation of vegetation is closely related to livestock grazing and excessive farmland reclamation. In the short term, overgrazing and the reclamation of cultivated land will reduce the biomass and species diversity of grassland plant communities and have a negative impact on the restoration of ecological communities [52]. Yu et al. [28], studying the impact of cultivated land expansion on drought in the Songnen Plain, also showed that the expansion of the cultivated land area destroys vegetation communities, increases the rate of grassland degradation, and indirectly affects the regional climate [53]. This trend slowed down after the implementation of the policy of returning farmland to forest and grassland in Northeast China in 2010, and the area of sandy land also decreased. In general, the area of vegetation coverage is still decreasing year by year. In addition, Ma et al. [54] concluded that vegetation degradation inevitably leads to significant increases in bare soil area and desert-like landscapes, whereas increases in vegetation area, farmland restoration, and natural succession may reverse desertification [55]. In arid and semi-arid areas, lakes are important geomorphological features and play a major role in maintaining the ecological balance of sandy land [56]. From 1980 to 2022, the lake area of the DBSL decreased, and some vegetation areas near the lake group also experienced varying degrees of degradation. Guo et al. [14] studied the interaction between vegetation area and water area changes near Taitema Lake and concluded that vegetation was highly sensitive to water area changes, with a significant decrease in vegetation area in dry years [57]. After the implementation of the ecological water conveyance project, water inflow into the lake increased significantly, and the vegetation area near the lake has generally increased. This is consistent with the results of this study. When the water around the lake is sufficient, vegetation growth is facilitated. Zhang et al. showed that the growth of high-density vegetation in sandy land can maintain soil mois-

ture, increase groundwater storage, and promote lake restoration [58–60]. On the whole, the vegetation areas in the DBSL are concentrated, and vegetation, lakes, and sandy land are scattered in a mosaic pattern. The shrinkage of the lake area leads to a decrease in the area of the surrounding vegetation coverage area year by year, which affects the interception of the vegetation on the nearby surface sand material, and the area of the sand landscape in the shrinking position of the lake area increases. The change in the vegetation landscape is related to the distribution of water resources in sandy land. The degree of vegetation coverage can change the mode of the surface wind–sand movement and affect the spatial distribution of sandy land [61].

### 4.2. Driving Mechanism of Vegetation–Lake–Sand Landscape Evolution

The evolution of the vegetation–lake–sand landscape in the study area is directly affected by annual average precipitation, annual average temperature, and annual average wind speed [62]. Increases in precipitation, temperature, wind speed, and potential evaporation lead to an increase in the surface sediment transport rate and sand area [63]. Water evaporation and lake area decrease, vegetation habitat is reduced, plant growth is inhibited, and the vegetation coverage area is reduced. Vegetated areas are sensitive to precipitation [64], and the change in the vegetation landscape follows the change trend of precipitation. In a study on the response of vegetation coverage to precipitation changes in typical sandy land in eastern China, Kaymaz et al. [65] confirmed this point [66].

In the present study, there was a spatial correlation between vegetation and annual average precipitation changes in the sandy land in the northwest of the DBSL. From 1995 to 2000, the annual average precipitation in the study area increased, and the high vegetation coverage area in the region gradually increased. The regional pattern of vegetation distribution during this period was mainly driven by precipitation. This is consistent with the results of Liu et al. [52] from their study on the sensitivity of sandy vegetation to water storage in arid and semi-arid areas of China [67]. The results of this study show that meteorological factors such as annual average temperature, annual precipitation, and annual average wind speed were closely related to the trend of lake area reduction in the past 42 years. The average annual evaporation of the DBSL increased slowly, the average annual temperature increased, and the average annual precipitation fluctuated. The sandy lakes were sensitive to climatic factors and had poor stability. The area of the DBSL lakes showed an increasing–decreasing–slowing and continuous decreasing trend from 1980 to 2022, which is consistent with the results obtained by Chen et al. [68]. Ji et al. [69] analyzed the change law of vegetation in the wind–sand area by combining climate and social factors and found that precipitation is the main climatic factor affecting vegetation change, but the quantitative results show that human activities are an important driving force for vegetation change [70–72]. With an increasing population, the demand for food is also increasing, and the area of urban construction land is expanding. In the 1990s, the desertification of the DBSL land intensified, large areas of vegetation were destroyed, and the lake area decreased sharply. Rapid urbanization and urban land expansion under economic development have led to the destruction of vegetation coverage, which is reflected in the long-term continuous degradation and short-term rapid attenuation of vegetation coverage caused by human encroachment on cultivated land, deforestation, and environmental damage. In the 2000s, after the implementation of the Three North Shelter-belt Development Plan and the policy of returning farmland to grassland and forest, the vegetation in the region gradually recovered, surface roughness increased, wind erosion weakened, desertification slowed down, and the ecological environment recovered [73]. The implementation of policies such as lake expansion and humidification and the returning farmland to wetland in the 2010s has also effectively slowed down the degradation of lakes and vegetation landscapes [74].

*4.3. DBSL Ecological Environment Management and Optimization Strategy*

The DBSL vegetation landscape is mainly distributed in Hulun Buir City and Ongniud Banner of Inner Mongolia. The ecological policy of the region is decreasing, so the original environmental policy should be maintained. The region should continue to implement the establishment of ecological forest areas; grasp the structure of agriculture, forestry, and animal husbandry; and rationally develop the forest, grassland, and other resources in the region to promote the restoration of the grassland ecosystem. The restoration of forests and grassland can change the characteristics of the soil surface, reduce wind erosion, and increase surface roughness. Planting different vegetation types can reduce wind erosion and enhance sand fixation [75]. The lake landscape in the sandy land is mainly distributed in the southwestern six banners of Hulun Buir City, the Horqin Left Wing Rear Banner of Tongliao City, and Baicheng City. The region is characterized by substantial human activities, leading to the deterioration of the ecological environment [76]. Therefore, the government should implement ecological projects such as returning farmland to forests, grasslands, and lakes to maintain the stability of the ecological environment [77]. The sandy landscape is mainly distributed in the southern part of the Hailar River in the eastern part of the Northeast Plain and the New Barag Left Banner, most of Jilin Province, and the northwestern part of Liaoning Province. This is mainly due to the economic development of the region. The proportion of the primary industry is substantial, but the corresponding ecological environment governance policies are lagging behind [78–80]. The awareness of the general public regarding environmental protection should be strengthened, and new environmental policies should be formulated and implemented. It should be pointed out that to restore environmental functions and promote the development of an ecological economy in Northeast China, ecological restoration measures should be adjusted according to the local conditions and ecological principles, combined with the characteristics and mechanisms of different landscape changes, to achieve the coordinated development of the environment and society [81].

**5. Conclusions**

The DBSL vegetation landscape is mainly distributed in the middle, northwest, and southwest of the region, covering an area of 30,508.82 km$^2$. The vegetation landscape in Northeast China is dominated by high and medium vegetation coverage areas. The DBSL lake landscape is concentrated in the center of the region, with small lake groups being scattered in the middle and lower parts to the southwest, and the total lake area covers 2667.554 km$^2$. The sandy landscape is mainly distributed in the south of the study area, and a small part is distributed in the central part of the study area. The sandy landscape covers an area of 10,546.07 km$^2$.

In the past 42 years, the vegetation–lake–sand landscape of the DBSL showed a dynamic change trend. The vegetation coverage area decreased by 85,46 km$^2$, among which, the high vegetation coverage area decreased significantly by 35.36% and the medium and low vegetation coverage areas decreased by 19.16% and 11.56%, respectively. The lake area showed a decreasing trend, and some lake groups shrank or dried up; the total lake area decreased by 1211.86 km$^2$. The sandy land area showed an overall increasing trend; from 2005 to 2010, the sandy land area increased sharply by 2254.892 km$^2$, an increase of 16.25%; after 2010, desertification decreased. There was an obvious interaction between vegetation–lake–sand landscape in Northeast China.

The analysis of the driving mechanism of vegetation–lake–sand landscape change in the northeast sandy land found that the human factor is the first main component, with a contribution rate of 54.45%, which is the main factor affecting the change of vegetation–lake–sand landscape in the research area. Since the population increase in the northeast region, the disturbance due to human activities has increased, which has caused a lot of negative effects on the ecological environment. As the second main component, natural factors have a contribution rate of 18.92%. To a certain extent, they are involved in the evolution of sandy and lake landscapes. The climate of Northeast China is

characterized by alternating wet and dry fluctuating changes, which, to a certain extent, influence the evolution of the vegetation–lake–sand landscape in the DBSL.

**Author Contributions:** Data curation, investigation, methodology, visualization, software, and writing—original draft, W.L.; supervision, validation, and formal analysis, G.T.; conceptualization and writing—review and editing, H.D. All authors have read and agreed to the published version of the manuscript.

**Funding:** This research was funded by the National Natural Science Foundation of China (no. 42271005).

**Institutional Review Board Statement:** Not applicable.

**Informed Consent Statement:** Informed consent was obtained from all subjects involved in the study.

**Data Availability Statement:** The data presented in this study are available upon request from the corresponding author.

**Conflicts of Interest:** The authors declare no conflicts of interest. The funders had no role in the collection, analysis, or interpretation of the data; in the writing of the manuscript; or in the decision to publish the results.

## Appendix A

**Table A1.** Table of landscape pattern change driving factors.

| Driving Factors | Factors | Description |
| --- | --- | --- |
| Natural factors | Annual average temperature | Spatial data |
| | Annual precipitation | Spatial data |
| | Annual average wind speed | Non-spatial data |
| | Annual average evaporation | Non-spatial data |
| Humanistic factors | Total population | Non-spatial data |
| | GDP | Non-spatial data |
| | Farmers' income | Non-spatial data |
| | Number of large livestock at the end of the year | Non-spatial data |
| | Arable land area | Non-spatial data |
| | Grain yield | Non-spatial data |
| | Feature value | Non-spatial data |
| | Contribution rate | Non-spatial data |
| | Cumulative contribution rate | Non-spatial data |

**Table A2.** PCA loading matrix of DBSL.

| Factors | PC1 | PC2 | PC3 |
| --- | --- | --- | --- |
| Annual average temperature | −0.28 | 0.10 | −0.30 |
| Annual precipitation | 0.41 | 0.57 | 0.17 |
| Annual average wind speed | −0.10 | 0.18 | −0.17 |
| Annual average evaporation | −0.03 | −0.60 | 0.57 |
| Total population | 0.16 | 0.35 | 0.65 |
| GDP | 0.78 | −0.30 | −0.30 |
| Farmers' income | 0.13 | −0.25 | 0.05 |
| Number of large livestock at the end of the year | 0.30 | 0.02 | −0.10 |
| Arable land area | −0.03 | −0.02 | 0.03 |
| Grain yield | 0.01 | 0.02 | −0.01 |
| Feature value | 5.03 | 1.74 | 0.96 |
| Contribution rate | 54.45 | 18.92 | 10.67 |
| Cumulative contribution rate | 54.45 | 73.37 | 84.04 |

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
