# Peer review of "Vegetation–Lake–Sand Landscape of Northeast China Sandy Land between 1980 and 2022: Pattern, Evolution, and Driving Forces"

_sustainability, doi:10.3390/su16083382_

Round 1

Reviewer 1 Report

Comments and Suggestions for Authors

This article studies a topic of great relevance and interest in the scientific field, focusing on the analysis of the vegetation, lake and sand landscape of northeastern China (it should be noted that it would have been of interest to incorporate the study of the year 2023). 

The work includes, following a coherent logical order of structure, the methodology of analysis (which is of notable originality and, above all, useful for the assessment and interpretation of the reality under study), the objectives pursued and the documentary basis required for its interpretation.

It would, however, be of interest to expand the analysis of the graphic material included in the study, giving greater relevance to the results obtained and their critical evaluation; otherwise, the work loses notoriety of the aforementioned reality. For this reason, it is recommended to carry out a more in-depth analysis of the data and information collected in the tables and maps that are incorporated in the work, as well as to interrelate them with each other, preventing them from being mere decorative elements, without linking each other to the others, in the interpretation of the aforementioned reality.

Likewise, it is recommended to improve the quality of the location map and include the “source” of maps and tables (although it is understood that they are of our own creation, but it would be of interest to specify this).

Reviewer 2 Report

Comments and Suggestions for Authors

Line 25. You just talked about the drivers and the most important thing is the socioeconomic factors, and I think it's important to be clear how does the socioeconomic drive it.

Line 67. What methods did others use to do the research, where and what the results were, and on a global scale, not just who did the research and summarized the similarities and differences.

Line 92. Your introduction explains the underlying mechanism, but it's not reflected in your abstract.

Line 118. In the study area overview map, you should put a large map of your study area, which can clearly see the plains, deserts, rivers, etc.

Line 157. I think you should make a chart of the general situation of the indicators you choose here, which can better reflect their general situation.

Line 266. The change of its spatio-temporal pattern has been shown in section 3.2.1, why explain it again here.

Line 309. The author is simply talking about correlations, not the internal relationships, such as how one indicator changes with another. Two unrelated data columns will also show a correlation when placed in the data analysis, so the correlation coefficient does not reflect the internal relationship.

Line 390. As before, the first part of the analysis is not logical and the summary is not clear enough. Second, you only show the correlation, you don't show the relationship.

Line 542. The summary of the conclusion is not clear enough, according to the purpose of your research, only the results.

Reviewer 3 Report

Comments and Suggestions for Authors

Summary: The article deals an interesting are of eco-socio environment. The manuscript contains the most up-to-date and relevant information about the given topic and written in a fully understandable, coherent way. Contains numerous amout of figures and one table. The manuscript is well written, but contains a lot of misspellings and typos that must corrected by the authros. Use the accepted SI metrics.

General comments:

- Title should be changes to cover more the upcoming topic of the manuscript. Eg.: Vegetation-Lake-Sand Landscape of Northeast China Sandy Land between 1980 and 2022: Pattern, Evolution, and Driving Forces

- Please describe what does the DBSL means, there is not a single way to find out the exact meaning of this abreviation.

- Aims are not in sync with the conclusion. The authors did not give any further utilization perspectives in the conclusion, but they aimed it during the study. Give advice and conclude based on your data.

- In line 146, I do not believe that only one settlement located in the given area.

- line 148-149: what is the percentage of the missing data compared to the overall data?

- line 338-339: Something is missing at the end of the sentence, without that I do not understand it. Please correct it.

Specific comments:

1. Statistical analysis and descreption of the used methods are totally missing, must added.

2. Figure 1 is hard to understand, what are the colors represents and the whole map is small. A table containing the data numerically be added next to the existing figure. Way more informative representation of the data.

3. Marking of sub-maps in Fig 3 are not in sync , during the description of them it is a good idea to numerically expressing the changes rates too, not just visually.

4. Fig. 4. must be made bigger with thicker coloumns and re-thingk the axis names (at this moment area/km2 means nothing). Make it to represent %, total coverage/km2, etc.

5. Fig 6. too much information, stick with only the presented years data and do not include more info.

6. Fig. 7: same here, to much resolution on the information, and why the 3rd sub-figure is named "a" and not the 1st one?

7. Table 1. should be moved to supplementary.

8. Capitalize vegetation, there were no "*" or "**" on the graph, so it means taht non of the data presented are significant?

Based on my advices, please correct the manuscript.

Comments on the Quality of English Language

English is fine, but a lot of typos and misspelling making it hard to understand.

Round 2

Reviewer 2 Report

Comments and Suggestions for Authors

Figure 1 also needs to be provided with a review number from the Ministry of Natural Resources of China.

The results section needs to be improved and the authors simply need to describe the results.

Comments on the Quality of English Language

Minor editing of English language required
